# Unifying Back-Propagation and Forward-Forward Algorithms through Model Predictive Control

## Abstract

We introduce a Model Predictive Control (MPC) framework for training deep neural networks, systematically unifying the Back-Propagation (BP) and Forward-Forward (FF) algorithms. At the same time, it gives rise to a range of intermediate training algorithms with varying look-forward horizons, leading to a performance-efficiency trade-off. We perform a precise analysis of this trade-off on a deep linear network, where the qualitative conclusions carry over to general networks. Based on our analysis, we propose a principled method to choose the optimization horizon based on given objectives and model specifications. Numerical results on various models and tasks demonstrate the versatility of our method.

## 1 Introduction

Neural Networks (NN) are rapidly developing in recent years and have found widespread application across various fields. While Back-Propagation (BP) (Rumelhart et al.) stands as the predominant training method, its high memory requirements limit its application to deep model, large batch sizes, and memory-constrained devices, such as those encountered in large language models (Touvron et al., 2023; OpenAI et al., 2023; Gu & Dao, 2023). To address these limitations, recent research has sought to mitigate the drawbacks associated with BP or explore alternative training methods altogether. Notably, Hinton (2022) proposed a Forward-Forward (FF) algorithm that uses layer-wise local loss to avoid backward propagation through layers. Xiong et al. (2020) also proposed a local loss learning strategy. Local loss methods are theoretically more memory-efficient and biologically realizable than BP. However, the performance of FF algorithms is often inferior to that of BP, particularly in deep models. Moreover, the mechanism of FF is unclear, and lacks understanding of why and when FF can work.

Is there an optimization method that can balance the memory usage and accuracy? Inspired by the parallels between Back-Propagation and the Pontryagin Maximum Principle (Li et al., 2017), we discover the similarity between local loss and the greedy algorithm. Utilizing the concept of the Model Predictive Control (MPC), we introduce the MPC framework for deep learning. In this framework, FF and BP are firstly unified, representing two extremes under this framework. Some other previous works (Xiong et al., 2020; Nøkland & Eidnes, 2019; Belilovsky et al., 2019) (Remark 3.1) are also incorporated in this framework. This framework offers not only a spectrum of optimization algorithms with varying horizons to balance performance and memory demand but also a dynamical viewpoint to understand the FF algorithm. Additionally, our theoretical analysis of the deep linear neural network shows that the gradient estimation converges polynomially as the horizon approaches full back-propagation (Theorem 3.4), leading to diminishing returns for sufficiently large horizons. However, the memory demand grows constantly with respect to the horizon, indicating an intermediate region offering a favorable trade-off between memory and accuracy. Based on this analysis, we propose horizon selection algorithms for different objectives to balance the accuracy-efficiency trade-off.

The contributions of this paper are summarized as follows:

- We propose a novel MPC framework for deep neural network training, unifying BP and FF algorithms and providing a range of optimization algorithms with different accuracy-memory balances.

- We analyze the accuracy-efficiency trade-off within the MPC framework, providing theoretical insights into gradient estimations and memory usage.

- We propose an objective-based horizon selection algorithm based on the previous theoretical result that gives the optimal horizon under the given objective.

- The theoretical finding and the horizon selection algorithm are validated under various models, tasks, and objectives, illustrating the efficacy of our MPC framework.

The remainder of this paper is structured as follows: Section 2 provides a review of relevant literature. In Section 3, we briefly review the BP and FF algorithms, and then present our proposed framework together with its analysis. Section 4 introduces the proposed horizon selection algorithm based on the previous theoretical analysis. The results of numerical experiments are shown in Section 5. Finally, we conclude the paper and discuss avenues for future research in Section 6.

## 2   LITERATURE REVIEW

Despite its great success in deep learning, it has been argued that Back-Propagation is memory inefficient and biologically implausible (Hinton, 2022). Recently, numerous studies have been focused on addressing the shortcomings of BP, such as direct feedback alignment algorithm (Nøkland, 2016), synthetic gradient (Jaderberg et al., 2017), and LoCo algorithm (Xiong et al., 2020). Notably, Hinton (2022) proposed a forward-forward algorithm that uses layer-wise local loss to avoid backward propagation through layers. Although these methods mitigate the memory inefficiency of BP, they often suffer from inferior performance or introduce additional structures to the models and the theoretical understanding of when and why these methods work is insufficient.

The connection and comparison between residual neural networks and control systems were observed by E. Subsequently, Li et al. (2017) used relationships between Back-Propagation and the Pontryagin maximum principle in optimal control theory to develop alternative training algorithms. Following this, several works have introduced control methods aimed at enhancing optimization algorithms in neural network training (Li et al., 2017; Weng et al.; Nguyen et al., 2024). These prior works have primarily relied on powerful PID controllers or optimal control theory, which, in practice, are rooted in the concept of backward propagation and thus inherit the drawbacks associated with BP. Conversely, MPC (Grüne & Pannek), another well-known controller, has yet to be extensively explored in the realm of deep learning (Weinan et al.).

The analysis of deep linear networks has been extensively explored in the literature like (Cohen et al., 2022; Arora et al., 2018b). Furthermore, research has investigated the convergence rates of coarse gradients in various contexts, such as quantifying neural networks (Long et al.), zeroth-order optimization methods (Chen et al., 2023; Zhang et al., 2024), and truncated gradients in recurrent neural networks (Aicher et al., 2019). Our work builds upon these analyses combine insights from both domains to analyze the performance of the proposed MPC framework on deep linear networks.

## 3   THE MODEL PREDICTIVE CONTROL FRAMEWORK

This section offers a brief overview of the traditional Back-Propagation (BP) algorithm and the novel Forward-Forward (FF) algorithm. We then establish connections between BP and FF algorithms using the MPC concept, introducing an MPC framework for deep learning models. Finally, we give a theoretical analysis of the MPC framework on linear neural networks, providing insights into the influence of horizon.

In the paper, we focus on the deep feed-forward neural networks, which have the following form:

$$x(t+1) = f_t(x(t), u(t)), \tag{1}$$

where $t \in \{0, ..., T-1\}$ denotes the block index, $x(t) \in \mathbb{R}^{n_t}$ denotes the input of the $t$-th block, and $u(t) \in \mathbb{R}^{m_t}$ represents vectorized trainable parameters in the $t$-th block, $n_t, m_t$ are dimensions of $x(t)$ and $u(t)$, $f_t : \mathbb{R}^{n_t} \times \mathbb{R}^{m_t} \to \mathbb{R}^{n_{t+1}}$ denotes the forward mapping of the $t$-th block. In deep learning training, the primary objective is to minimize the empirical loss using gradient:

$$J(u) = L(x(T)), \quad u^{\tau+1} = u^\tau - \eta g(u^\tau), \tag{2}$$

where $u \triangleq (u(0)^\top, \cdots, u(T-1)^\top)^\top \in \mathbb{R}^m$ denotes all trainable parameters in the neural network, $m = \sum_{t=0}^{T-1} m_t$ is the number of trainable parameters in the model and $g(u)$ represents the gradient, $\eta$ denotes the learning rate, and $L(x(T)) : \mathbb{R}^{n_T} \to \mathbb{R}$ represents the loss on the final output $x(T)$[1], $\tau$ denotes the iteration index and. In this paper, we further assume the loss to be compatible with the state $x(t)$ for all $t = 1, \cdots, T$, i.e. either $x(t)$ maintains the same dimension for all $t = 1, \cdots, T$ or there is a linear projection (e.g. Pooling layer) that unifies $x(t)$ to the same dimension in the loss.

## 3.1 BACK-PROPAGATION AND FORWARD-FORWARD ALGORITHM

The Back-Propagation algorithm (Rumelhart et al.) computes the gradient of the loss function for the weights of the neural network:

$$g_{\text{BP}}(u(t)) = \nabla_{u(t)} J(u) = \nabla_{u(t)} L(x(T)). \tag{3}$$

Recently, Hinton (2022) proposed the Forward-Forward (FF) algorithm as an alternative to BP. The FF algorithm computes gradients using the local loss function of the current block:

$$g_{\text{FF}}(u(t)) = \nabla_{u(t)} L(x(t+1)). \tag{4}$$

In the original paper (Hinton, 2022), the author used the Euclidean norm of the layer output as a loss, but here we use the more general form of the loss function.

While the FF algorithm is more memory-efficient than BP, its performance may be worse due to the lack of global information. In contrast, BP is typically more accurate but demands more memory. Moreover, the theoretical understanding of when and why FF might perform well remains limited. Leveraging the concept of MPC in deep learning models, we propose a novel MPC framework that unifies the BP and FF algorithms. This framework offers a spectrum of optimization algorithms with varying horizons, allowing for a balance between performance and memory demand.

## 3.2 ADOPTING MPC FOR DEEP LEARNING

In this section, we introduce the classical Model Predictive Control (MPC) (Grüne & Pannek) and explore its connection with deep learning models. Subsequently, we propose an MPC framework tailored to deep learning models and demonstrate how it unifies BP and FF algorithms within this framework

MPC is an optimization-based method for the feedback control of dynamic systems. In the infinite horizon control problem, which is commonly considered in the control field, the trajectory loss is used, i.e.:

$$J(u) = \sum_{t=0}^{\infty} l(t, x(t), u(t)), \tag{5}$$

where $l(t, \cdot, \cdot) : \mathbb{R}^{n_t} \times \mathbb{R}^{m_t} \to \mathbb{R}$ is the trajectory loss for state $x(t)$ and control $u(t)$ on time $t$. Due to the computational complexity of obtaining optimal control for an infinite time horizon, a truncated finite-time control problem is solved at each time step:

$$J^h(t, z, u_t^h) = \sum_{s=t}^{t+h-1} l(s, x(s), u(s)) \tag{6}$$

$$\text{s.t.} \quad x(t) = z, x(s+1) = f_s(x(s), u(s)), \forall s = t, \cdots, t+h-1,$$

where $h$ is the horizon considered by MPC at each time step, $u_t^h \triangleq \{u(t), \cdots, u(t+h-1)\}$. The solution $u_t^{h*} = \operatorname{argmin}_{u_t^h} J^h(t, x(t), u_t^h)$ is applied to the current state:

$$x(t+1) = f_t(x(t), u_t^{h*}(t)). \tag{7}$$

To apply the concept of MPC to deep learning models, we consider the deep learning model as a dynamic system governed by the underlying dynamic function $f_t$ (1), where the block index $t$

---

[1]Here, the loss is the average of loss in training dataset, we ignore the data index for brevity. We also ignore regularization terms and other possible parameters and labels in the loss function, focusing solely on the loss function that depends on the final output $x(T)$.

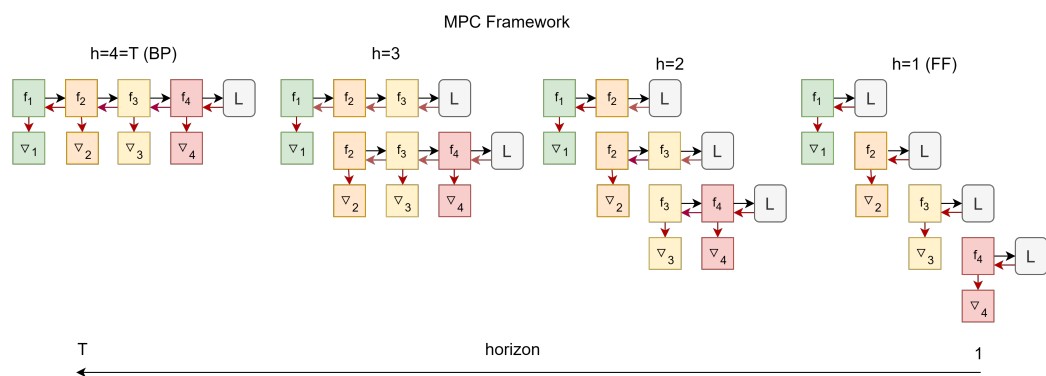

Figure 1: Diagram of MPC framework on a 4-block model: black arrows denote the forward pass and red arrows denote the backward pass, $\nabla_t \triangleq g_h(u(t))$ is the gradient of $t$-th block. MPC uses partial gradient propagation. We can see that BP can be seen as MPC with the full horizon ($h = T$), while FF is MPC with horizon 1 ($h = 1$)

corresponds to the time index in control systems. The weights $u(t)$ and input $x(t)$ of the $t$-th block are analogous to the control and state at time $t$ (E; Li et al., 2017).

Since deep learning problems typically involve only a terminal loss applied at the end of the network, we need to define an effective trajectory loss $l(t, x(t), u(t))$ for each block $t$ that add up to the desired terminal loss (2). One particular choice is as follows:

$$l(t, x(t), u(t)) \triangleq L(f_t(x(t), u(t))) - L(x(t)) = L(x(t+1)) - L(x(t)). \tag{8}$$

We demonstrate that the terminal loss $L(x(T))$ is equivalent to the sum of trajectory loss $\sum_{t=0}^{T-1} l(t, x(t), u(t))$ up to a constant (see Appendix C.1).

Using a similar definition of $J^h$ in (6), we propose the MPC framework for deep learning models in the form of:

$$g_h(u(t)) = \nabla_{u(t)} J^h(t, x(t), u_t^h), \tag{9}$$

where $u_t^h = \{u(t), ..., u(\min(t + h - 1, T))\}$

Instead of evaluating the entire model as BP or only the local loss of the current block as FF, MPC considers a horizon $h$ from the present block. Moreover, as illustrated in Figure 1, MPC provides a family of training algorithms with different horizons $h$, and both the FF and BP algorithms can be seen as extreme cases of the MPC framework with horizon $h = 1$ and $h = T$ respectively,

$$g_{\text{FF}}(u(t)) = g_1(u(t)), \quad g_{\text{BP}}(u(t)) = g_T(u(t)). \tag{10}$$

A detailed example is provided in the Appendix A for a better understanding of the MPC framework.

**Remark 3.1** *Other algorithms using local loss (e.g. (Nøkland & Eidnes, 2019; Belilovsky et al., 2019)) can also be seen as MPC framework with horizon $h = 1$. Moreover, LoCo algorithm (Xiong et al., 2020) can be seen as the MPC framework with horizon $h = 2$ under larger blocks (each stage in the ResNet-50 be seen as a block, refer to Appendix B).*

**Remark 3.2** *Noted that $L(x(t))$ in the (8) is independent of $u(t)$, and $J^h(t, x(t), u_t^h) = \sum_{s=t}^{t+h-1} l(s, x(s), u(s)) = L(x(t+h)) - L(x(t))$ where only $L(x(t+h))$ depends on $u(t)$. Therefore, in practice, there is no need to compute $L(x(t))$ for $J^h$*

The MPC framework, akin to the FF algorithm, is advantageous in terms of memory usage since it considers only part of the model in the computation of $J^h$. In practice, the memory demand for training a deep learning model largely depend on the need to store intermediate value for Back-Propagation, theoretically proportional to the depth of the model. Therefore, the memory usage will be of linear growth in horizon $h$, i.e.:

$$M(h) = ah + b, \tag{11}$$

where $M(h)$ is memory usage for horizon $h$, and $a$, $b$ are independent of $h^2$. Numerical experiments also verify the linear dependency of the horizon (Appendix E.1).

Empirically, the MPC framework realizes the trade-off between performance and memory demand by providing a spectrum of optimization algorithms with varying horizons $h$. A larger horizon provides an accurate gradient, while a smaller horizon gives memory efficiency. This idea helps to understand the validity of the FF algorithm and its reduced accuracy compared to BP. However, there is still a lack of theoretical analysis on the influence of horizon $h$ on accuracy. In the following sections, we provide theoretical results on the influence of horizon $h$ on deep linear networks and propose a horizon selection algorithm for the given objective.

**Remark 3.3** *In the traditional control field, finite time problems are rarely studied especially for the MPC method. The existing results on the performance of MPC are primarily either asymptotic or rely on properties that are challenging to verify or not suitable for deep learning. For a more detailed introduction to traditional MPC methods, interested readers may refer to Grüne & Pannek.*

### 3.3 THEORETICAL ANALYSIS ON DEEP LINEAR NETWORKS

From the previous discussion, we conjectured that a larger horizon will give better performance. In this section, we theoretically investigate the influence of horizon $h$ on linear neural networks.

Considering a simple Linear NN problem with linear fully connected layers and quadratic loss, which is widely used in the theoretical analysis of deep learning (Arora et al., 2018b; Cohen et al., 2022). The parameter $u(t)$ is the weight $W(t)$ in the $t$-th layer, omitting bias for brevity:

$$f_t(x(t), u(t)) = W(t)x(t), \ L(x(T)) = \frac{1}{2}\|x(T) - y\|_2^2, \tag{12}$$

where $(x(0), y) \in \mathbb{R}^n \times \mathbb{R}^n$ is one sample input-label pair in the dataset, $x(t) \in \mathbb{R}^n$ has the same dimension for all $0 \leq t \leq T$, $W(t) \in \mathbb{R}^{n \times n}$.

We aim to analyze the deviation between the gradient obtained by the MPC framework $g_h$ and the true gradient $g_T$. Considering that the difference in the scale of the gradient norm can be compensated by adjusting the learning rate, the minimum deviation between the rescaled gradient $g_h$ and true gradient $g_T$ can be determined by the angle between these two vectors.

$$\min_{c_h} \|c_h g_h - g_T\|_2 = \left\| \frac{\|g_T\|_2}{\|g_h\|_2} \cos(\theta_h) g_h - g_T \right\|_2 = \sin(\theta_h)\|g_T\|_2, \tag{13}$$

where $\theta_h = \arccos(\frac{g_h^\top g_T}{\|g_h\|_2\|g_T\|_2})$ denotes the angle between $g_h$ and $g_T$.

We further assume the dataset is whitened, i.e. the empirical (uncentered) covariance matrix for input data $\{x(0)\}$ is equal to identity as (Arora et al., 2018a). In this case, the problem is equivalent to $L(u) = \frac{1}{2}\|W - \Phi\|_F^2$, where $W \triangleq W(T-1)\cdots W(0)$, $\Phi$ is the empirical (uncentered) cross-covariance matrix between inputs $\{x(0)\}$ and labels $\{y\}$. We have the following result for the asymptotic gradient deviation in the deep linear network at Gaussian initialization.

**Theorem 3.4 ((Informal) Gradient Deviation in Deep Linear Network)** *Let* $W(t) = I + \frac{1}{T}\tilde{W}(t)$, $\{\tilde{W}(t)\}$ *are matrices with bounded 2-norm, i.e.* $\exists c > 0$ *such that* $\|\tilde{W}(t)\|_2 \leq c, \forall t$. *Denote* $\theta_h$ *the angle between* $g_h$ *and* $g_T$. *When* $T \to \infty, h \to \infty, \frac{h}{T} = \alpha, 1 - \cos^2(\theta_h) = O((1 - \frac{h}{T})^3)$ *as* $\frac{h}{T} \to 1$.

The proof estimate the norm of gradient to bound $\cos(\theta_h)$ for $\frac{h}{T} \to 1$. The complete statement and the proof is in Appendix C.2. The polynomial relationship between $\cos(\theta_h)$ and $h$ is also observed for nonlinear cases (refer to Section 5.1). Further using the previous study of biased gradient descent, we can get the linear convergence to a non-vanishing right-hand side for strong convex loss with rate $O(\cos^2(\theta_h))$ (see Appendix C.3), i.e. the loss decrease speed is linear with $\cos^2(\theta_h)$:

$$r(h) = \frac{\ln(J_h(\tau)/J_0)}{\ln(J_T(\tau)/J_0)} = O(\cos^2(\theta_h)), \tag{14}$$

---

[2]Influence of other factors like batch size are assumed to be constant and absorbed in coefficients $a$ and $b$.

Theorem 3.4 indicates that the gradient deviation between $g_h$ and $g_T$ aligns with the growth of the horizon $h$, leading to an improved performance, which verifies the previous conjecture. However, the performance gain diminishes cubically as the horizon approaches the full horizon $T$, suggesting that excessively increasing the horizon may not bring significant performance improvements. Nevertheless, memory usage continues to increase linearly with the horizon, indicating a non-trivial optimal horizon in the middle. When horizon approach 1, the numerical experiments show that $\cos(\theta_h)$ has non-fixed value and non-zero derivative (Section 5.1). This observation underscores the importance of selecting an optimal horizon to balance accuracy and memory efficiency, as illustrated in the following example.

For instance, if we weigh the performance and cost linearly with memory usage, i.e. $C(M(h)) = M(h) = ah + b$, where $M(h)$ is the memory usage for horizon $h$. Assuming $\cos^2(\theta_h) = 1 - c(1 - \frac{h}{T})^3$ and target objective is $-r(h) + \lambda C(M(h))$ where $a, b, c > 0$. If $\frac{3c}{aT^2} < \lambda < \frac{3c(T-2)^2}{aT^2}$, we can get the optimal horizon $h^* = T(1 - \sqrt{\frac{a\lambda}{3c}})$ and $2 \le h^* \le T - 1$, the optimal horizon will neither be 1 (FF) nor $T$ (BP).

This example illustrates that the optimal horizon depends on various factors such as the model, dataset, task, and objective. In the subsequent section, we will provide horizon selection algorithms for different objectives to navigate the accuracy-efficiency trade-off.

# 4 OBJECTIVE-BASED HORIZON SELECTION ALGORITHM OF MPC FRAMEWORK

From the previous section, we know the optimal horizon will depend on both the deep learning problem and objective function. Utilizing the above theoretical results, in this section we propose a horizon selection algorithm for the given objective, aimed at achieving the balance between accuracy and memory efficiency.

There are a variety types of objectives that consider accuracy and efficiency. In this paper, we consider objectives that are functions of the relative rate of loss decrement $r(h)$ (14), and the memory-depend cost $C(M)$. From the analysis in Section 3.3, we may use $\cos(\theta_h)$ to approximate $r(h)$. As for cosine similarity $\cos(\theta_h)$ and memory usage $M(h)$, these properties are hard to analyze but easy to compute and fit using polynomial fitting with low degree (order 3 for $\cos(\theta_h)$ and order 1 for $M(h)$) on just a few horizons and batches of data based on (11) and Theorem 3.4, i.e.

$$\cos(\hat{\theta}_h) = \text{Polyfit}(H, \{\cos(\theta_h)\}_{h \in H}, order = 3)(h) \tag{15}$$

$$\hat{M}(h) = \text{Polyfit}(H, \{M(h)\}_{h \in H}, order = 1)(h). \tag{16}$$

When it comes to the cost $C(M)$, the practical significance of this consideration is that memory usage is related to the cost on the device in practice[3]. Though any cost function is applicable, for simplicity we consider following two kinds of cost functions: 1) Linear cost function $C(M) = c\frac{M}{M_0}$, where $M_0$ is the memory of one node (e.g. GPU) and $c$ is the unit cost, corresponding to the ideal case where computing resources can be divided unlimited; 2) Ladder cost function $C(M) = c\lceil\frac{M}{M_0}\rceil$, reflecting a more realistic situation where only an integer number of GPUs are allowed.

Since the limited memory resource case will be trivial and we just need to select the largest horizon under the memory limitation, we consider the following two objectives that are more general and practical: accuracy constraint and weighted objective.

**Objective 1: Accuracy Constraint**

$$\max_{h \in H} \quad C(M(h))$$
$$\text{s.t.} \quad r(h) \ge 1 - \epsilon. \tag{17}$$

In this case, we want to get a good performance using minimum cost. The algorithm selects the smallest horizon that can meet the accuracy constraint using the proposed loss estimation.

---

[3]The popular cloud platforms now all provide pay-per-use mode which the pricing is mainly linear with the memory of GPUs, like Google Cloud `https://cloud.google.com/vertex-ai/pricing#text-data` and Huawei Cloud `https://www.huaweicloud.com/intl/en-us/pricing/#/ecs`

**Objective 2: Weighted Objective** We can also consider the case when there is no hard constraint, but the objective is a weighted sum of performance and memory efficiency:

$$\max_{h \in H} \quad -r(h) + \lambda C(M(h)). \tag{18}$$

The horizon selection algorithm (Algorithm 1) first compute the true cosine similarity $\cos(\theta_h)$ and memory usage $M(h)$ in the subset $H$, and fit for other horizon by polynomial; then the optimal horizon $h^{*\,4}$ for the given objective is solved using a traditional optimization algorithm (e.g. brute force search).

---

**Algorithm 1** Horizon Selection Algorithm

---

**Require:** Objective $O(r, c)$, Cost function $C(M)$, Dataset D, Subset of horizon $H$, (Optional)
1: **for** $h$ in $H$ **do**
2:    $M(h) \leftarrow$ memory usage for horizon $h$
3: **end for**
4: **for** $h \in \{1, \cdots, T\}$ **do**
5:    $\hat{M}(h) \leftarrow \text{Polyfit}(H, \{M(h)\}_{h \in H}, order = 1)(h)$ (Eq. (16))
6: **end for**
7: **for** $h$ in $H$ **do**
8:    **for** batch $b$ in D **do**
9:       $g_{h,b} \leftarrow \nabla_u J^h$ using Eq. (9) and Lemma C.1 on batch $b$
10:    **end for**
11:    $\cos(\theta_h) \leftarrow \mathbb{E}_{[b \text{ in D}]} \frac{g_{h,b}^{\top} g_{T,b}}{\|g_{h,b}\|_2 \|g_{T,b}\|_2}$
12: **end for**
13: **for** $h \in \{1, \cdots, T\}$ **do**
14:    $\cos(\hat{\theta_h}) \leftarrow \text{Polyfit}(H, \{\cos(\theta_h)\}_{h \in H}, order = 3)(h)$ (Eq. (15))
15: **end for**
16: $\hat{r}(h) \leftarrow \cos^2(\hat{\theta_h})$ (Eq. (14))
17: $\hat{h}^* \leftarrow \text{argmin}_h O(\hat{r}(h), C(\hat{M}(h)))$
18: **return** $\hat{h}^*$

---

## 5 EXPERIMENT RESULTS

To evaluate the effectiveness of the proposed MPC framework for deep learning models, we conduct some distinct experiments: 1) linear residual Neural Network (Res Linear NN); 2) residual MLP (Res MLP); 3) 62-layer ResNet model (Sec 4.2 (He et al., 2016)) (ResNet-62); 4) fine-tuning ResNet-50 (He et al., 2016) and ViT-b16 (Dosovitskiy et al., 2021). Detailed model structure, dataset, and training settings can be found in Appendix D, and all the experiments are conducted in NVIDIA GeForce RTX 3090 using TensorFlow (Abadi et al., 2016).

In summary, our findings reveal that:

1. The theoretical result of the linear NN model is qualitatively consistent with numerical experiments on nonlinear models with respect to polynomial convergence of gradient and the linear growth of memory usage (Section 5.1).

2. The performance of larger horizons will be better and will converge to the results of BP algorithm (Full horizon) quickly (Section 5.2).

3. The optimal horizon depends on the task, model structure, and objective. Both BP and FF algorithms can be optimal horizons, but in most cases, an intermediate horizon will be the best choice. The proposed horizon selection algorithm can help select the horizon throughout many cases, especially for difficult tasks (Section 5.3).

These experiments collectively demonstrate that the proposed MPC framework is applicable across various deep learning models and tasks, particularly for large-scale deep learning models where memory demands during training are significant.

---

[4]If Eq. (11) and Eq. (14) are correct, the returned horizon will be optimal.

## 5.1 QUALITATIVE VERIFICATION FOR POLYNOMIAL CONVERGENCE OF $g_h$ AND MEMORY USAGE

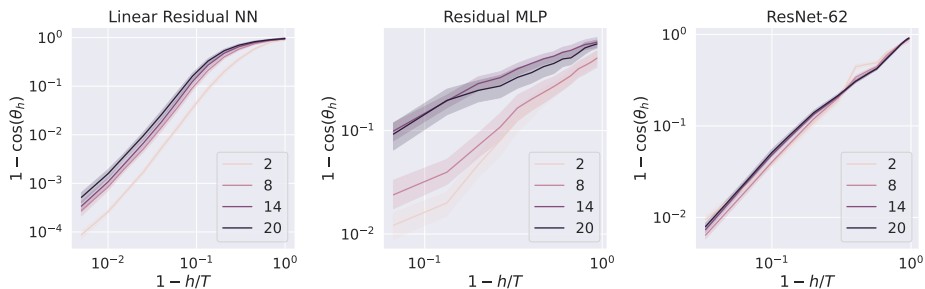

Figure 2: Relationship Between $g_h$ and $h$ on different models.
The x-axis shows $T - h$ and y-axis shows $1 - \cos(\theta_h)$. Each line represents a different training epoch. **Left:** Linear residual NN, **Middle:** Residual MLP, **Right:** ResNet-62

To verify the angle $\theta_h$ between $g_h$ and $g_T$ for different horizons, we trained a series of neural networks on the first three tasks. The results are depicted in Figure 2. The results demonstrate that the gradient convergence rate is polynomial for all models throughout the entire training process, consistent with our theoretical analysis, despite that the order might not be as high as observed in simple linear models. Furthermore, the angle of the same horizon tends to increase during training, suggesting that a small horizon can be efficient in the early training epochs but not in the later training epochs.

Further, in Section 3.3, we have argued that the memory usage is linear with respect to the horizon $h$. To verify this, we trained both the classic ResNet-50 model He et al. (2016) and the Vision Transformer (ViT) model Dosovitskiy et al. (2021) on CIFAR100 dataset. We use the same batch size to train the models in different horizons. Under the hypothesis that the memory usage is linear with respect to the horizon, the batch size should be reciprocal to the horizon, and the results depicted in Figure 3 demonstrate it. Since the memory usage is halved after a downsampling operation in the resnet-50 model (the width and height are halved while the number of channel is doubled) to maintain the flops constant, the memory usage is not linear in the resnet-50 model but still increasing with horizon being larger. However, for the ViT model, the memory usage is linear as each transformer block has the same size. Further discussion on the memory usage can be found in Appendix E.1.

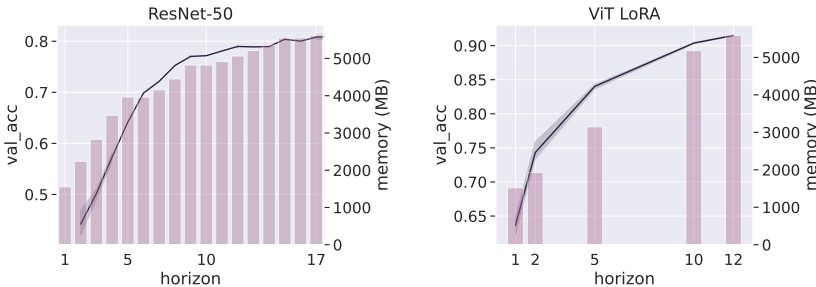

Figure 3: Test accuracy and memory usage of full tuning ResNet-50 and LoRA tuning ViT-b16 on CIFAR100. Dark line shows the loss of final epoch and shallow bars shows the memory usage of the horizon. The maximum
**Left:** ResNet-50, **Right:** ViT-b16

## 5.2 PERFORMANCE OF DIFFERENT HORIZONS

From Figure 3, we can also observe that for large horizons, the performance will converge to the full horizon (BP) case, which is consistent with the convergence of the gradient (Thm 3.4) and indicates

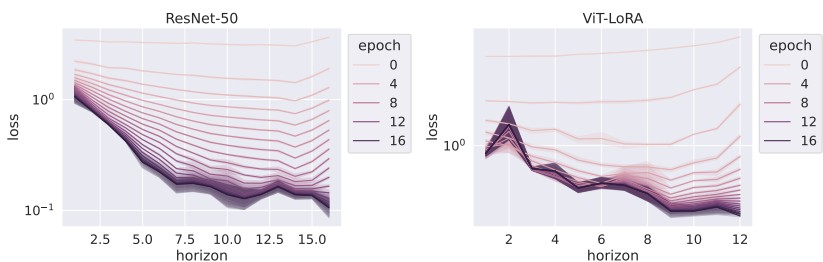

Figure 4: Full tuning ResNet-50 and LoRA tuning ViT-b16 with different horizons
**Left:** ResNet-50, **Right:** ViT-b16

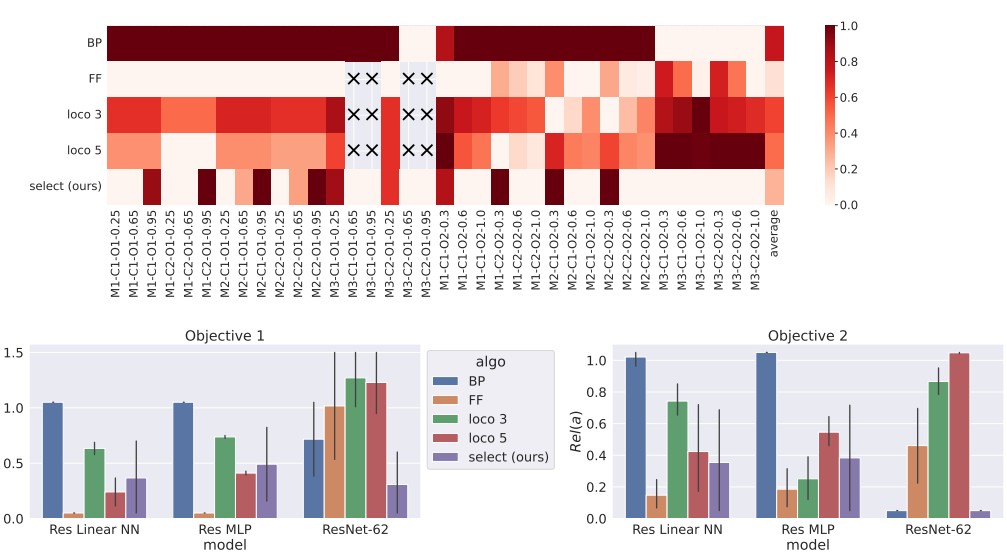

Figure 5: Relative performance $Rel(a)$ of different algorithms on various models, costs, and objectives. **Up:** Heatmap of relative objective value of baselines and horizon selection algorithms and objectives, y-axis: different algorithms, x-axis: models, tasks, and objectives (refer to Table D3), color: relative objective value, cross notation '$\times$': infeasible solution. **Bottom:** Average relative performance of different algorithms on different models (the infeasible solution is treated as 1.5)

the possibility of optimality in the intermediate horizon. The results reveal that the performance of the MPC framework is highly dependent on the horizon. We further show the detailed snapshots in different training procedure in Figure 4. We find that, in the early training epochs, the loss decrement is aligned for different horizons. However, as training progresses, the loss decrement of different horizons diverges especially for small horizons. The convergence of large horizons to BP algorithm is also observed. Further results of the performance of different horizons on the first three tasks can be found in Appendix E.2.

## 5.3 VALIDATION OF HORIZON SELECTION ALGORITHM

**Relative Performance**  Since the objective value varies from different models, tasks, costs, and objective parameters, we transform the results to relative performance for comparison. Denoting $O(a)$ be the objective value using algorithm $a$, the relative performance is calculated by mapping the best result $\min_a O(a)$ to the worst result $\max_a O(a)$ onto $[0, 1]$, i.e.

$$Rel(a) = \frac{O(a) - \min_a O(a)}{\max_a O(a) - \min_a O(a)}. \tag{19}$$

To demonstrate the effectiveness of the proposed horizon selection algorithm, we trained neural networks on the first three tasks using three selection algorithms and tested its objectives with the

baseline (BP and FF algorithm and LoCo algorithm). The results, depicted in Figure 5, demonstrate that our proposed algorithms can achieve better objective values compared to BP. Since FF utilizes only one block each time, its memory usage will be minimal, leading to its relatively low objective value, especially in objective 1, while it also might not be able to satisfy the constraints. As for the LoCo algorithm, which can still be seen as a fixed horizon algorithm, it might also violate the constraints and is less flexible among different models and tasks. As illustrated in Figure 5 our proposed horizon selection algorithm is more efficient than BP and more stable than FF and gets comparable average performance for the LoCo algorithms. We also find that the horizon selection algorithm gets better results on difficult tasks, e.g. the training of ResNet-62. Since the inaccuracy of loss decreases speed estimation, the horizon selection algorithm tends to select a relatively larger horizon to satisfy the constraint, thus it gets poor results in Objective 1 compared to 2.

## 6 DISCUSSION

This study presents a systematic framework that integrates the Forward-Forward algorithm with Back-Propagation, offering a family of algorithms to train deep neural networks. Theoretical analysis on deep linear networks demonstrates a polynomial convergence of gradients concerning the horizon and the diminished return for large horizons. Based on the analysis result and various objectives, we propose an objective-based horizon selection algorithm to balance performance and memory efficiency. Additionally, numerical experiments across various tasks verify the qualitative alignment of theory and underscore the significant impact of horizon selection. However, many phenomena in the MPC framework are not fully analyzed or understood. Here are some limitations of this study.

Firstly, the analysis is primarily based on deep linear networks, but analyzing nonlinear deep networks remains challenging, and existing results heavily rely on the BP algorithm and are thus not applicable to the proposed framework. Nevertheless, numerical experiments show that our theoretical result qualitatively complies with the behavior observed in nonlinear models and the analysis provides enough insight for the horizon selection algorithm.

Secondly, while the numerical result demonstrates the significant potential of the MPC framework, our proposed horizon selection algorithm involves simplifications and empirical fittings. However, loss estimation is not our primary focus, and the experiments show that the proposed algorithm achieves a good balance between performance and memory efficiency compared to baselines. Future studies could refine this algorithm with more precise loss estimation techniques.

Thirdly, our research focuses solely on the horizon's influence, neglecting other MPC framework hyperparameters like block selection and loss splitting methods which are also important. We leave these for the future studies.

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

## A  AN EXAMPLE OF THE MPC FRAMEWORK

In this section, we give a detailed example of 5-layer residual MLP with MSE loss and $h = 1, 3, 5(= T)$ to illustrate the MPC framework.

**Structure**  Each layer of the network is a fully-connected layer with activation and residual connection. We take each layer to be a block in the MPC framework, i.e. $T = 5$.

$$f_t(x(t), u(t)) = x(t) + \sigma(Linear(x(t); u(t))), \forall t = 0, \cdots, 4,$$

$$L(x) = MSE(x, y)$$

Note that we keep $x$ in loss $L$ and ignore other possible parameters and labels.

**Trajectory loss**  Trajectory loss $l(t, x(t), u(t))$ is defined as loss decrement between block input $x(t)$ and block output $x(t + 1)$:

$$l(t, x(t), u(t)) = L(x(t+1)) - L(x(t)) = MSE(f_t(x(t), u(t)), y) - MSE(x(t), y), \forall t = 0, \cdots, 4.$$

**Horizon $h = 3$ case**  The truncated loss $J^h(t, x(t), u_t^h)$ is defined as partial sum of the trajectory loss.

$$J^3(0, x(0), u_0^3) = \sum_{t=0}^{2} l(t, x(t), u(t)) = L(x(3)) - L(x(0))$$

$$= MSE(f_2(\cdot, u(2)) \circ f_1(\cdot, u(1)) \circ f_0(\cdot, u(0))(x(0)), y) - MSE(x(0), y)$$

where $u_0^3 = \{u(0), u(1), u(2)\}$, and $\circ$ means composition of functions in argument $\cdot$, i.e. $g(\cdot) \circ f(x, \cdot)(y) \triangleq g(f(x, y))$. Similarly,

$$J^3(1, x(1), u_1^3) = L(x(4)) - L(x(1)), \ J^3(2, x(2), u_2^3) = L(x(5)) - L(x(2)),$$

$$J^3(3, x(3), u_3^3) = L(x(5)) - L(x(3)), \ J^3(4, x(4), u_4^3) = L(x(5)) - L(x(4)).$$

The gradient is defined as

$$g_3(u(0)) = \nabla_{u(0)} J^3(0, x(0), u_0^3) = \nabla_{u(0)} L(x(3)),$$

Second equality is because $x(0)$ is independent with $u(0)$, thus $\nabla_{u(0)} L(x(0)) = 0$. Similarly,

$$g_3(u(1)) = \nabla_{u(1)} L(x(4)), \ g_3(u(2)) = \nabla_{u(2)} L(x(5)),$$

$$g_3(u(3)) = \nabla_{u(3)} L(x(5)), \ g_3(u(4)) = \nabla_{u(4)} L(x(5)).$$

**Horizon $h = 1$ (FF) case**  When $h = 1$, the truncated loss $J^h(t, x(t), u_t^h)$ is the current trajectory loss:

$$J^1(t, x(t), u_t^1) = l(t, x(t), u(t)) = L(x(t + 1)) - L(x(t)), \forall t = 0, \cdots, 4,$$

and thus

$$g_1(u(t)) = \nabla_{u(t)} J^1(t, x(t), u_t^1) = \nabla_{u(t)} L(f_t(x(t), u(t))), \forall t = 0, \cdots, 4,$$

which is the gradient of a local loss as the Forward-Forward algorithm.

**Horizon $h = 5 = T$ (BP) case**  When $h = T$, the truncated loss $J^h(t, x(t), u_t^h)$ is the sum of the following trajectory loss:

$$J^T(t, x(t), u_t^T) = \sum_{s=t}^{T-1} l(s, x(s), u(s)) = L(x(T)) - L(x(t)), \forall t = 0, \cdots, 4,$$

and

$$g_T(u(t)) = \nabla_{u(t)} L(x(T)), \forall t = 0, \cdots, 4,$$

which has the same gradient as traditional Back-Propagation.

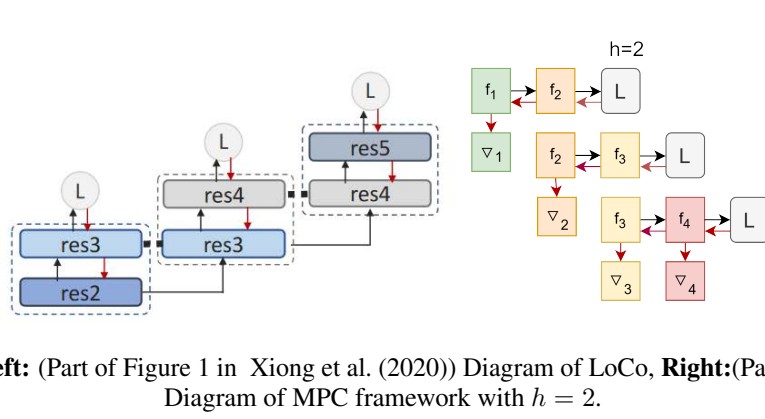

Figure B1: **Left:** (Part of Figure 1 in Xiong et al. (2020)) Diagram of LoCo, **Right:**(Part of Figure 1) Diagram of MPC framework with $h = 2$.

## B  COMPARISON OF LoCo AND MPC FRAMEWORK WITH $h = 2$

In this section, we demonstrates that LoCo is included in the MPC framework with $h = 2$. As shown in Figure B1, we can find that the structure of LoCo is identical with the MPC framework with $h = 2$ (refer to Figure 1) if we rotate their diagram clockwise for 90 degrees. The only difference is that LoCo computes the gradient of the intermediate stage twice in each step, i.e.

$$g_{LoCo}(u(t)) = \nabla_{u(t)} L(x(t+1)) + \nabla_{u(t)} L(x(t+2)), \forall t = 1, \cdots, t-2. \tag{20}$$

This can be realized in the MPC framework by changing trajectory loss to:

$$l(t, x(t), u(t)) = \begin{cases} 0, & t = 0 \\ L(x(t+1)), & t > 0 \end{cases} \tag{21}$$

However, this change leads to the sum of trajectory loss not equal to the terminal loss, i.e. $J^T(t, x(t), u_t^T) \neq L(x(T))$. For the consistency of the paper, we use the "split loss" for the LoCo structure in our paper.

## C  PROOF OF THEOREM

### C.1  EQUIVALENCE OF TERMINAL LOSS AND TRAJECTORY LOSS

In this section, we prove the equivalence of terminal loss and trajectory loss. We use $J_{tra}(u)$ to denote the trajectory loss and $J_{ter}(u)$ to denote the terminal loss in the section to distinguish.

**Trajectory Loss to Terminal Loss**  To transform trajectory loss to terminal loss, we need to add an auxiliary state $x_0$ and let the auxiliary state $x_0$ record the accumulation of trajectory loss and $x_0(T)$ be considered as the terminal loss:

$$x_0(t+1) = x_0(t) + l(t, x(t), u(t)), x_0(0) = 0 \tag{22}$$
$$J_{ter}(u) = L(x(T), u) = x_0(T). \tag{23}$$

It is easy to prove using induction on $x_0(t)$ that

$$J_{ter}(u) = x_0(T) = \sum_{t=0}^{T-1} l(t, x(t), u(t)) = J_{tra}(u). \tag{24}$$

**Terminal Loss to Trajectory Loss**  On the other hand, to turn terminal loss to trajectory loss, we use split loss (8) $l(t, x(t), u(t)) \triangleq L(x(t+1)) - L(x(t))$. The terminal loss will become the accumulation of loss decreases for each block, and the equivalence can be proven by simple calculation:

$$\begin{aligned} J_{tra}(u) &= \sum_{t=0}^{T-1} l(t, x(t), u(t)) \\ &= \sum_{t=1}^{T} (L(x(t)) - L(x(t-1))) \\ &= L(x(T)) - L(x(0)) = J_{ter}(u) - L(x(0)). \end{aligned} \tag{25}$$

Since $L(x(0))$ only depends on input and is unrelated to the model, it is a constant for the model.

## C.2 PROOF OF THEOREM 3.4

We formally state the Theorem 3.4 and prove it here.

**Theorem 3.4 (Formal) Gradient Deviation in Deep Linear Network** *Let* $W(t) = I + \frac{1}{T}\tilde{W}(t)$, $\{\tilde{W}(t)\}$ *are matrices with bounded 2-norm, i.e.* $\exists c > 0$ *such that* $\|\tilde{W}(t)\|_2 \leq c$ *for all t. Denote* $\theta_h$ *the angle between* $g_h$ *and* $g_T$. *Then* $\exists C_1, C_2 > 0$ *such that:*

$$\lim_{\alpha \to 1} \liminf_{T \to \infty, h = \alpha T} \frac{1 - \cos^2(\theta_h)}{(1-\alpha)^3} = C_1,$$

$$\lim_{\alpha \to 1} \limsup_{T \to \infty, h = \alpha T} \frac{1 - \cos^2(\theta_h)}{(1-\alpha)^3} = C_2$$

(26)

To simplify notation, denoting that $W_{t_1}^{t_2} \triangleq W(t_2 - 1)W(t_2 - 2) \cdots W(t_1), 0 \leq t_1 < t_2 \leq T$. We can get $\frac{dL}{dW^t} = (W_0^t - \Phi)$ for the given constant $\Phi$ being the covariance matrix of $x, y$. The gradient of control $u(t) = W(t)$ for horizon $h$ can be derived as:

$$\frac{dL_h}{du(t)} = \begin{cases} (W_{t+1}^{t+h})^\top \frac{dL}{dW^{t+h}}(W_0^{t-1})^\top, & 0 \leq t < T - h \\ (W_{t+1}^T)^\top \frac{dL}{dW^T}(W_0^{t-1})^\top, & T - h \leq t < T, \end{cases}$$

(27)

In the following, we omit the round down for $W_{t_1}^{t_2}$ and $W(t)$, i.e. $W_{\alpha T}^{\beta T} \triangleq W_{\lfloor \alpha T \rfloor}^{\lfloor \beta T \rfloor}$ and $W(\alpha T) \triangleq W(\lfloor \alpha T \rfloor) \, \forall 0 \leq \alpha \leq \beta \leq 1$.

To prove the theorem, we use the following lemmas:

**Lemma C.1** *Under the condition in Theorem 3.4,* $\exists - \infty < \lambda_1 \leq \lambda_2 < \infty$, *such that* $\forall 0 \leq \beta \leq \gamma \leq 1$:

$$\|W_{\beta T}^{\gamma T}\|_F \geq \sigma_{\min}(W_{\beta T}^{\gamma T}) \geq e^{(\gamma - \beta)\lambda_1}, \quad \|W_{\beta T}^{\gamma T}\|_F \leq e^{(\gamma - \beta)\lambda_2}.$$

(28)

and

**Lemma C.2** *Under the condition in Theorem 3.4,* $\exists \lambda_1, \lambda_2, y_1, y_2 \in \mathbb{R}$, *such that* $\lambda_1 \leq \lambda_2, \forall \beta \in (0, 1 - \alpha]$

$$\left\|\frac{dL_{\alpha T}}{du(\beta T)}\right\|_F \geq (e^{(\alpha + \beta)\lambda_1} - y_1)e^{(\alpha + \beta)\lambda_1}$$

$$\left\|\frac{dL_{\alpha T}}{du(\beta T)}\right\|_F \leq (e^{(\alpha + \beta)\lambda_2} - y_2)e^{(\alpha + \beta)\lambda_2}.$$

(29)

**Lemma C.3** *Under the condition in Theorem 3.4, and the same* $\lambda_1, \lambda_2, y_1, y_2 \in \mathbb{R}$ *as in Lemma C.2,* $\forall \beta \in (1 - \alpha, 1]$

$$\left\|\frac{dL_{\alpha T}}{du(\beta T)}\right\|_F \geq (e^{\lambda_1} - y_1)e^{\lambda_1}$$

$$\left\|\frac{dL_{\alpha T}}{du(\beta T)}\right\|_F \leq (e^{\lambda_2} - y_2)e^{\lambda_2}.$$

(30)

**Lemma C.4** *Under the condition in Theorem 3.4,* $\forall \epsilon > 0$, $\exists \alpha_0(\epsilon) > 0$, *such that* $\forall \alpha > \alpha_0(\epsilon), T > 0$,

$$\|I - W_{\alpha T}^T\|_F \leq \epsilon.$$

(31)

**Lemma C.5** *For arbitrary fixed constant* $c, y \in \mathbb{R}$, *denoting* $g_{\alpha T}(\cdot; c, y)$ *a function of* $\beta$ *in* $[0, 1]$:

$$g_{\alpha T}(\beta; c, y) = \begin{cases} e^{c(\alpha + \beta)}(e^{c(\alpha + \beta)} - y), & 0 < \beta < 1 - \alpha \\ e^c(e^c - y), & 1 - \alpha \leq \beta \leq 1. \end{cases}$$

(32)

*and then define $f(\cdot; c, y) : \mathbb{R} \to \mathbb{R}$*

$$f(\alpha; c, y) \triangleq \frac{\int_0^1 g_{\alpha T}(\beta; c, y) g_T(\beta; c, y) \, d\beta}{\sqrt{\int_0^1 g_{\alpha T}^2(\beta; c, y) \, d\beta} \sqrt{\int_0^1 g_T^2(\beta; c, y) \, d\beta}}. \tag{33}$$

*Then:*

$$f^2(\alpha; c, y) = 1 + O((1 - \alpha)^3), \ as \ \alpha \to 1. \tag{34}$$

The proof of these lemmas is postponed after the proof of the Theorem 3.4. Assuming we have all the lemmas above, we can prove Theorem 3.4

**Proof of Theorem 3.4** First we need to get the bound of the cosine similarity $\cos(\theta_h)$. The upper bound of $\cos(\theta_h)$ can be derived by the Frobenius norm:

$$\begin{aligned}
\cos(\theta_h) &= \frac{\sum_{t=0}^{T-1} \text{tr}((\frac{dL_h}{du(t)})^\top \frac{dL_T}{du(t)})}{\sqrt{\sum_{t=0}^{T-1} \|\frac{dL_h}{du(t)}\|_F^2} \sqrt{\sum_{t=0}^{T-1} \|\frac{dL_T}{du(t)}\|_F^2}} \\
&\leq \frac{\sum_{t=0}^{T-1} \|\frac{dL_h}{du(t)}\|_F \|\frac{dL_T}{du(t)}\|_F}{\sqrt{\sum_{t=0}^{T-1} \|\frac{dL_h}{du(t)}\|_F^2} \sqrt{\sum_{t=0}^{T-1} \|\frac{dL_T}{du(t)}\|_F^2}}.
\end{aligned} \tag{35}$$

As for the lower bound, since $\frac{dL_h}{du(t)}$ and $\frac{dL_T}{du(t)}$ will align when $\frac{h}{T} = \alpha \to 1$, we can still use rescaled Frobenius norm to get the lower bound.

For any $c_1 \in [0, 1]$, denoting $\lambda_2$ the larger one in LemmaC.1 and Lemma C.2 and $\lambda_1$ be the smaller one, using Lemma C.4 we have $\forall \epsilon > 0, \exists \alpha_0(\epsilon) > 0$, such that $\forall \alpha > \alpha_0(\epsilon), T > 0$,

$$\|I - W_{\alpha T}^T\|_F \leq \epsilon. \tag{36}$$

Combined with Lemma C.1 we can get: $\forall 0 \leq \beta \leq 1 - \alpha$

$$\|W_{\beta T}^{(\alpha+\beta)T} - W_{\beta T}^T\|_F \leq \|W_{\beta T}^{(\alpha+\beta)T}(I - W_{(\alpha+\beta)T}^T)\|_F \leq \|W_{\beta T}^{(\alpha+\beta)T}\|_F \|I - W_{(\alpha+\beta)T}^T\|_F \leq \epsilon e^{\alpha \lambda_2}, \tag{37}$$

$$\left\| \frac{dL}{dW^{(\alpha+\beta)T}} - \frac{dL}{dW^T} \right\|_F = \|W^{(\alpha+\beta)T}(I - W_{(\alpha+\beta)T}^T)\|_F \leq \epsilon e^{\lambda_2}. \tag{38}$$

Then using Lemma C.2, (37) and (38)

$$\begin{aligned}
&\left\| \frac{dL_{\alpha T}}{du(\beta T)} - \frac{dL_T}{du(\beta T)} \right\|_F \\
&\leq \|W^{\beta T}\|_F \left( \|W_{\beta T}^{(\alpha+\beta)T}\|_F \left\| \frac{dL}{dW^{(\alpha+\beta)T}} - \frac{dL}{dW^T} \right\|_F + \left\| \frac{dL}{dW^T} \right\|_F \|W_{\beta T}^{(\alpha+\beta)T} - W_{\beta T}^T\|_F \right) \\
&\leq e^{(\alpha+\beta+1)\lambda_2}(1 + e^{\lambda_2} - y_2)\epsilon \leq e^{2\lambda_2}(1 + e^{\lambda_2} - y_2)\epsilon,
\end{aligned} \tag{39}$$

let $\epsilon = \sqrt{1 - c_1^2} e^{-2\lambda_2}(1 + e^{\lambda_2} - y_2)^{-1} \min_\gamma \{(e^{\gamma\lambda_1} - y_1)e^{\gamma\lambda_1}\} \leq \sqrt{1 - c^2} e^{-2\lambda_2}(1 + e^{\lambda_2} - y_2)^{-1} \|\frac{dL_T}{du(t)}\|_F$ we can get:

$$\left\| \frac{dL_{\alpha T}}{du(\beta T)} - \frac{dL_T}{du(\beta T)} \right\|_F \leq \sqrt{1 - c_1^2} \left\| \frac{dL_T}{du(\beta T)} \right\|_F, \tag{40}$$

thus

$$\left( vec\left( \frac{dL_{\alpha T}}{du(\beta T)} \right) \right)^\top vec\left( \frac{dL_T}{du(\beta T)} \right) \geq c_1 \left\| \frac{dL_{\alpha T}}{du(\beta T)} \right\|_F \left\| vec \frac{dL_T}{du(\beta T)} \right\|_F, \tag{41}$$

where $vec(\cdot)$ stands for matrix vectorization in column-first order. Eq. (41) is due to $\|vec(A)\|_2 = \|A\|_F$ and the following fact. For arbitrary two vector $a, b \in \mathbb{R}^n$, if $\|a - b\|_2 \leq \sqrt{1 - c_1^2}\|b\|_2$ then

$$\begin{aligned}
&\|a\|_2^2 - 2a^\top b + \|b\|_2^2 \leq 1 - c_1^2\|b\|_2^2 \\
&\Longrightarrow 2a^\top b \geq c_1^2\|b\|_2^2 + \|a\|_2^2 \\
&\Longrightarrow a^\top b \geq |c_1| \|a\|_2 \|b\|_2.
\end{aligned} \tag{42}$$

Then we have the lower bound for $\cos(\theta_{\alpha T})$ for all $\alpha > \alpha_0$ and uniformly in $T$:

$$\cos(\theta_{\alpha T}) \geq \frac{c_1 \sum_{t=0}^{T-1} \|\frac{dL_{\alpha T}}{du(t)}\|_F \|\frac{dL_T}{du(t)}\|_F}{\sqrt{\sum_{t=0}^{T-1} \|\frac{dL_{\alpha T}}{du(t)}\|_F^2} \sqrt{\sum_{t=0}^{T-1} \|\frac{dL_T}{du(t)}\|_F^2}}. \tag{43}$$

which again induce similar equation as (35) with additional scalor $c_1$.

Let $T \to \infty$, since we can multiply both numerator and denominator by $\frac{1}{T}$ and $\|\frac{dL_{\alpha T}}{du(t)}\|_F, \|\frac{dL_T}{du(t)}\|_F$ are bounded uniformly in $\alpha, T, t$, we have for all $\alpha \in (0, 1]$,

$$\lim_{T \to \infty} \left| \frac{\sum_{t=0}^{T-1} \|\frac{dL_{\alpha T}}{du(t)}\|_F \|\frac{dL_T}{du(t)}\|_F}{\sqrt{\sum_{t=0}^{T-1} \|\frac{dL_{\alpha T}}{du(t)}\|_F^2} \sqrt{\sum_{t=0}^{T-1} \|\frac{dL_T}{du(t)}\|_F^2}} - \frac{\int_0^1 \|\frac{dL_{\alpha T}}{du(\beta T)}\|_F \|\frac{dL_T}{du(\beta T)}\|_F \, d\beta}{\sqrt{\int_0^1 \|\frac{dL_{\alpha T}}{du(\beta T)}\|_F^2 \, d\beta} \sqrt{\int_0^1 \|\frac{dL_T}{du(\beta T)}\|_F^2 \, d\beta}} \right| = 0, \tag{44}$$

uniformly in $\alpha$.

Since the result of LemmaC.5 holds for arbitrary $c$ and $y$, $\exists C_1', C_2'$, such that

$$\lim_{\alpha \to 1} \liminf_{T \to \infty} \frac{1 - \cos^2(\theta_{\alpha T})}{(1-\alpha)^3} \geq \min_{\lambda \in [\lambda_1, \lambda_2], y \in [y_1, y_2]} \lim_{\alpha \to 1} \frac{1 - f^2(\alpha; \lambda, y)}{(1-\alpha)^3} = C_1',$$
$$\lim_{\alpha \to 1} \limsup_{T \to \infty} \frac{c_1 - \cos^2(\theta_{\alpha T})}{(1-\alpha)^3} \leq \max_{\lambda \in [\lambda_1, \lambda_2], y \in [y_1, y_2]} \lim_{\alpha \to 1} \frac{c_1(1 - f^2(\alpha; \lambda, y))}{(1-\alpha)^3} = C_2'. \tag{45}$$

The Compactness of $[\lambda_1, \lambda_2], [y_1, y_2]$ and the continuity of $g$ thus $f$ on $c$ and $y$ ensure the existence and bounded limit. Let $c \to 1$, we know that $\exists C_1, C_2$ such that

$$C_1' \leq C_1 = \lim_{\alpha \to 1} \liminf_{T \to \infty} \frac{1 - \cos^2(\theta_{\alpha T})}{(1-\alpha)^3} \leq \lim_{\alpha \to 1} \limsup_{T \to \infty} \frac{1 - \cos^2(\theta_{\alpha T})}{(1-\alpha)^3} = C_2 \leq C_2'. \tag{46}$$

$\square$

Below we prove the previous lemmas

**Proof of Lemma C.1:** Since $\{\tilde{W}(t)\}$ have bounded 2-norm $\|\tilde{W}(t)\|_2 \leq c$ for all $t$, the norm of the multiplication of $W(t)$ can be bounded, i.e. $\forall 0 \leq \beta \leq \gamma \leq 1$:

$$\|W_{\beta T}^{\gamma T}\|_2 \leq \Pi_{t=\beta T}^{\gamma T-1} \|W(t)\|_2 \leq (1 + \frac{c}{T})^{(\gamma-\beta)T} < e^{(\gamma-\beta)c}. \tag{47}$$

For the lower bound, since for arbitrary matrices $A, B \in \mathbb{R}^{n \times n}$ and non-zero vector $u \in \mathbb{R}^n$, the following inequality holds:

$$\|BAu\|_2 \geq \sigma_{\min}(B)\|Au\|_2 \geq \sigma_{\min}(B)\sigma_{\min}(A)\|u\|_2, \tag{48}$$

where $\sigma_{\min}(A)$ denotes the minimum singular value of matrix $A$, thus

$$\|BA\|_2 = \sup_{u \neq 0} \frac{\|BAu\|_2}{\|u\|_2} \geq \inf_{u \neq 0} \frac{\|BAu\|_2}{\|u\|_2} = \sigma_{\min}(BA) \geq \sigma_{\min}(B)\sigma_{\min}(A), \tag{49}$$

and we can get the following result:

$$\|W_{\beta T}^{\gamma T}\|_2 \geq \sigma_{\min}(W_{\beta T}^{\gamma T}) \geq \Pi_{t=\beta T}^{\gamma T-1} \sigma_{\min}(W(t)) \geq (1 - \frac{c}{T})^{(\gamma-\beta)T} > e^{-(\gamma-\beta)c}, \tag{50}$$

for all $0 \leq \beta \leq \gamma \leq 1$, . By equivalence between matrix norms, $\exists -\infty < \lambda_1 \leq \lambda_2 < \infty$ such that:

$$\|W_{\beta T}^{\gamma T}\|_F \geq e^{(\gamma-\beta)\lambda_1}, \|W_{\beta T}^{\gamma T}\|_F \leq e^{(\gamma-\beta)\lambda_2}. \tag{51}$$

hold for all $0 \leq \beta \leq \gamma \leq 1$ $\square$

**Proof of Lemma C.2** Given constant $\Phi$, we have:

$$\left|\|W^{t+h}\|_F - \|\Phi\|_F\right| \leq \|W^{t+h} - \Phi\|_F = \left\|\frac{dL}{dW^{t+h}}\right\|_F \leq \|W^{t+h}\|_F + \|\Phi\|_F \tag{52}$$

$$\sigma_{\min}\left(\frac{dL}{dW^{t+h}}\right) \geq \sigma_{\min}(W^{t+h}) - \sigma_{\max}(\Phi) \tag{53}$$

Eq. (53) can be derived from the following argument: for arbitrary matrix $A, B \in \mathbb{R}^{n \times n}$ and non-zero vector $u \in \mathbb{R}^n$:

$$\begin{aligned}
\sigma_{\min}(A - B) &= \min_{u \neq 0, \|u\|_2 = 1} \|(A - B)u\|_2 \\
&\geq \min_{u \neq 0, \|u\|_2 = 1} \|Au\|_2 - \max_{u \neq 0, \|u\|_2 = 1} \|Bu\|_2 \\
&= (\sigma_{\min}(A) - \sigma_{\max}(B)).
\end{aligned} \tag{54}$$

Denoting $\beta = \frac{t}{T}, \alpha = \frac{h}{T}$, from Lemma C.1, $\exists -\infty < \lambda_1 \leq \lambda_2 < \infty$, such that $\forall 0 \leq \beta \leq \gamma \leq 1$

$$\|W_{\beta T}^{\gamma T}\|_F \geq \sigma_{\min}(W_{\beta T}^{\gamma T}) \geq e^{(\gamma - \beta)\lambda_1}, \|W_{\beta T}^{\gamma T}\|_F \leq e^{(\gamma - \beta)\lambda_2}. \tag{55}$$

Combine (52), (53) and (55) let $\lambda_1 = \sigma_{\max}(\Phi), \lambda_2 = -\|\Phi\|_F$ we can get: $\forall \beta \in (0, 1]$

$$\begin{aligned}
\left\|\frac{dL_{\alpha T}}{du(\beta T)}\right\|_F &\geq \sigma_{\min}(W_t^{t+h})\sigma_{\min}\left(\frac{dL}{dW^{t+h}}\right)\sigma_{\min}(W_0^{t-1}) \geq (e^{(\alpha+\beta)\lambda_1} - y_1)e^{(\alpha+\beta)\lambda_1}, \\
\left\|\frac{dL_{\alpha T}}{du(\beta T)}\right\|_F &\leq \|W_t^{t+h}\|_F \|\frac{dL}{dW^{t+h}}\|_F \|W_0^{t-1}\|_F \leq (e^{(\alpha+\beta)\lambda_2} - y_2)e^{(\alpha+\beta)\lambda_2}.
\end{aligned} \tag{56}$$

$\square$

**Proof of Lemma C.3** Substitute $W^{t+h}, W_t^{t+h}$ in Lemma C.2 to $W^T, W_t^T$ and we can get the result.

**Proof of Lemma C.4** For arbitrary matrices $A, B \in \mathbb{R}^{n \times n}$ and arbitrary matrix norm $\|\cdot\|$

$$\|AB - I\| \leq \|AB - B\| + \|B - I\| \leq \|B\|\|A - I\| + \|B - I\|. \tag{57}$$

For arbitrary $\alpha \in [0, 1]$, substituting $A$ with $W_{\alpha T+1}^T$, $B$ with $W(\alpha T)$ and use 2-norm we have:

$$\|W_{\alpha T}^T - I\|_2 \leq \|W_{\alpha T+1}^T - I\|_2 \|W(\alpha T)\|_2 + \|W(\alpha T) - I\|_2 \leq \|W_{\alpha T+1}^T - I\|_2(1 + \frac{c}{T}) + \frac{c}{T}. \tag{58}$$

By induction we can derive that:

$$\|W_{\alpha T}^T - I\|_2 \leq \frac{c}{T} \frac{(1 + \frac{c}{T})^{(1-\alpha)T} - 1}{\frac{c}{T}} = (1 + \frac{c}{T})^{(1-\alpha)T} - 1 \leq e^{(1-\alpha)c} - 1, \tag{59}$$

then $\forall \epsilon$, let $\alpha_0 = 1 - \frac{\ln(1 + \frac{\epsilon}{\sqrt{n}})}{c}, \forall \alpha > \alpha_0$ we can get:

$$\|W_{\alpha T}^T - I\|_F \leq \sqrt{n}\|W_{\alpha T}^T - I\|_2 \leq \sqrt{n}(e^{(1-\alpha)c} - 1) \leq \sqrt{n}(e^{(1-\alpha_0)c} - 1) \leq \epsilon. \tag{60}$$

$\square$

**Lemma C.5** Since the result of LemmaC.5 can be derived from direct computation, we omit it.

**Remark C.6** *The result can be extended to any other random matrix ensemble as long as Lemma C.1 and Lemma C.2 holds.*

**Remark C.7** *Noted that the estimation of general inner product between $\frac{dL_h}{du(t)}$ and $\frac{dL_h}{du(t)}$ is hard, so we cannot get the expression for $\alpha \to 0$, while the numerical experiments show that the case of $\alpha \to 0$ is trivial, i.e. $\lim_{\alpha \to 0+} \cos(\theta_{\alpha T}) \neq 0$ and $\lim_{\alpha \to 0+} \frac{d\cos(\theta_{\alpha T})}{d\alpha} \neq 0$*

## C.3 Theory for Loss Decrease Speed

Once the cosine similarity and the norm of the rescaled gradient $g_h$ are specified, we can use the result of biased gradient descent (Theorem 4.6 Bottou et al. (2016) or Theorem 4 Chen & Luss (2018)) to get a linear convergence to a non-vanishing right-hand side for strong convex loss:

**Theorem C.8 (Theorem 4.6 Bottou et al. (2016))** *Assuming that $J(u)$ is c-strong convex, bounded below $J(u) \geq J(u^*)$, $\nabla J(u)$ is L-Lipschitz continuous, i.e. $\forall u, u' \in \mathbb{R}^m$*

$$J(u') \geq J(u) + \nabla J(u)^\top (u' - u) + \frac{c}{2}\|u - u'\|_2^2 \tag{61a}$$

$$\|\nabla J(u) - \nabla J(u')\|_2 \leq l\|u - u'\|_2. \tag{61b}$$

*Further assume that the stochastic gradient estimation $g(u;\xi)$ satisfies $\exists \mu_G \geq \mu \geq 0$ and $M \geq 0$, $M_V \geq 0$, such that $\forall \tau \in \mathbb{N}$*

$$\nabla J(u^\tau)^\top \mathbb{E}_\xi[g(u^\tau;\xi)] \geq \mu\|\nabla J(u^\tau)\|_2^2 \tag{62a}$$

$$\|\mathbb{E}_\xi[g(u^\tau;\xi)]\|_2 \leq \mu_G\|\nabla J(u^\tau)\|_2 \tag{62b}$$

$$\mathbb{V}_\xi[g(u^\tau;\xi)] \leq M + M_V\|\nabla J(u^\tau)\|_2^2. \tag{62c}$$

*Then for fixed learning rate $0 < \eta \leq \frac{\mu}{L(M_V + \mu_G^2)}$, the following inequality holds for all $\tau \in \mathbb{N}$*

$$J(u^\tau) - J(u^*) \leq (1 - \eta c\mu)^{\tau-1}(J(u^0) - J(u^*) - \frac{\eta LM}{2c\mu}) + \frac{\eta LM}{2c\mu}. \tag{63}$$

In our case, let the gradient estimator be the rescaled gradient for horizon $h$, i.e. $g(u) = g_h(u)$, then $\mu = \cos^2(\theta_h)$, so the linear decrease rate is $\ln(1 - \eta c\mu) = O(\mu) = O(\cos^2(\theta_h))$

## D Training Details

### D.1 Architecture

**Linear Residual NN and Residual MLP** 15-layer Fully connected neural networks with the same width 10 in each layer. Residual connection is applied to every layer except the first layer and last layer. For the linear residual NN, there is no activation function and bias in the fully connected layer. For the residual NNs, the activation function is ReLU and there is bias in the fully-connected layer.

**ResNet-62**  Each block has two convolution layers with batch normalization and activation. For each stage, before the first block, there will be a convolution layer with a specific stride to half the width and height of the feature. Since each convolution residual block has two convolution layers, the origin paper He et al. (2016) counts the number of convolution layers instead of the residual block.

Table D1: CNN structure

| Stage | Operator | #Chennels | #Blocks | Strides |
|-------|----------|-----------|---------|---------|
| stem | Input | 3 | - | - |
| 1 | ConvResBlock | 16 | 10 | 1 |
| 2 | ConvResBlock | 32 | 10 | 2 |
| 3 | ConvResBlock | 64 | 10 | 2 |

**LoCo 3 and LoCo 5**  As discussed in the Remark 3.1 and Appendix B, LoCo algorithm is a variant of the MPC framework with horizon 2 for larger blocks. In order to apply LoCo algorithm to the linear residual NN, residual MLP and ResNet-62, we implement two version of LoCo: LoCo 3 and LoCo 5, where the number indicates the total block of the model, and each block contains the same number of layers. For example, LoCo 5 has 5 blocks, and each block has 3 layers for 15-layer residual MLP, and for LoCo3 on ResNet-62, each stage is viewed as a block.

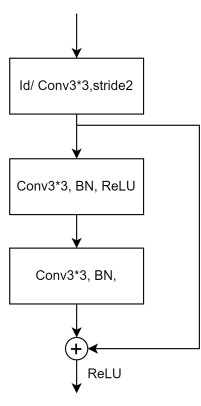

Figure D1: Diagram of ConvResBlockHe et al. (2016)

### D.2  TRAINING SETTINGS

**General training settings**  For all experiments, we employ SGD as the optimizer, and decrease the learning rate by 0.9 whenever the loss of the current epoch increases. We evaluate the performance using MSE loss for the first two experiments and cross-entropy loss and accuracy for the latter two experiments. The training settings are shown in Table D2.

Table D2: Training settings for all tasks

| Settings | Res Linear NN | Res MLP | ResNet | Fine-tuning |
|----------|---------------|---------|--------|-------------|
| Learning Rate | 0.03 | 0.01 | 0.01 | 0.001 |
| Batch Size | 100 | 100 | 32 | 64 |
| Epoch | 40 | 40 | 40 | 30 |
| Sample Size | 10000 | 100000 | 45000 | 50000 |
| Optimizer | SGD | SGD | SGD | Adam |
| Dataset | linear dataset[1] | trigonometric dataset[2] | CIFAR10 | CIFAR100 |

[1] Synthetic dataset, $y = W_0 x$ for Gaussian randomized matrix $W_0$, $w_{0,ij} \sim N(0, N^{-\frac{1}{2}})$, $x_i \sim N(0, N^{-\frac{1}{2}})$

[2] Synthetic dataset, $y = (1 + \epsilon) * (\cos(\pi x), \sin(\pi x), \cos(2\pi x), \sin(2\pi x))^\top$, $x \sim U([-2, 2]), \epsilon \sim N(0, 0.03)$

### D.3  NOTATIONS FOR FIGURE 5

Table D3 shows the notation used in the Figure 5

## E  FURTHER EXPERIMENTS

Here we supplement the two numerical experiments of the verification of the relationship memory $M(h)$ and horizon $h$, and the performance of different horizons in the first three experiments.

Table D3: Notations of Models, Costs, Objectives, and Parameters

| Model | | Cost[1] | | Objective | | |
|---|---|---|---|---|---|---|
| Notation | Meaning | Notation | Meaning | Notation | Meaning | Parameter |
| M1 | Linear Residual NN | C1 | $C(M) = c\frac{M}{M_0}$ | O1 | Objective 1 | $1 - \epsilon$ |
| M2 | Residual MLP | C2 | $C(M) = c\lceil \frac{M}{M_0} \rceil$ | O2 | Objective 2 | $\lambda$ |
| M3 | ResNet-62 | | | | | |

[1] In the experiment, we use $c = 1$ and $M_0 = \max_h\{M(h)\}$ for linear cost (C1) and $M_0 = 0.3 \max_h\{M(h)\}$ for ladder cost (C2).

## E.1 RELATIONSHIP BETWEEN MEMORY($h$) AND $h$

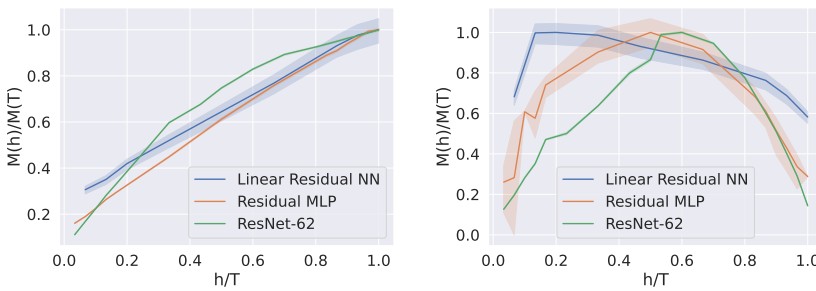

Figure E1: Relationship between $M(h)$ and $h$
x-axis is $\frac{h}{T}$, y-axis is memory usage ratio
**Left:** Memory usage for different horizon under eager mode, **Right:** Memory usage for different horizon under static mode

We also verify the memory usage on the first three tasks. In TensorFlow 2.0 or above versions, there are two modes: eager mode and static mode. The eager mode will release memory efficiently and give the theoretical linear dependency between memory usage and horizon, while the static mode will create the whole computation graph, causing every back-propagation to be counted, resulting in $O(h(T - h + 1))$ memory usage (refer to Figure 1). In the paper, we hypothesis that the model is running in the eager mode, i.e. the memory is efficiently used thus linear/affine with respect to horizon $h$.

## E.2 PERFORMANCE FOR DIFFERENT HORIZONS IN LINEAR RESIDUAL NN, RESIDUAL MLP, AND RESNET-62

Figure E2 illustrates the performance of different horizons on the three tasks. The results reveal that the performance of the MPC framework is highly dependent on the horizon. In the early training epochs, the loss decrease is aligned for different horizons, which is obvious in the residual NN case, indicating that the MPC framework can serve as an efficient warm-up training strategy. However, as training progresses, the loss decrease of different horizons diverges. Especially for small horizons, the loss might remain high, highlighting the importance of selecting the optimal horizon. These two findings are consistent with the deductions in the previous section. Furthermore, the comparison of the three tasks demonstrates that the performance of the same horizon varies across different models and tasks.

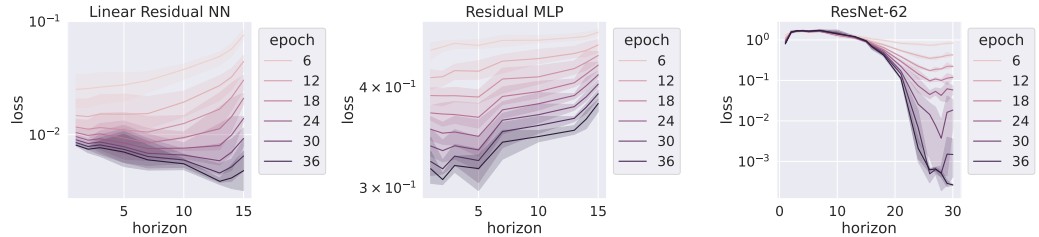

Figure E2: Training loss for different horizons
**Left:** Linear residual NN on linear regression, **Middle:** Residual MLP on trigonometric regression,
**Right:** ResNet-62 on CIFAR-10

