# OpenReview forum: "Unifying Back-Propagation and Forward-Forward Algorithms through Model Predictive Control"
_ICLR.cc/2025/Conference — ICLR 2025 Conference Withdrawn Submission_

### Official Review · Reviewer_Wf2p · 2024-10-29

**Soundness:** 2
**Presentation:** 2
**Contribution:** 2
**Rating:** 3
**Confidence:** 3

**Summary:**

This paper presents a training framework for deep neural networks formalized based on Model Predictive Control (MPC), where the horizon length can be adjusted to encompass both Back-Propagation (BP) and Forward-Forward (FF) algorithms as special cases. The framework allows for a flexible trade-off between memory usage and model performance by varying the horizon length. The authors provide an asymptotic theoretical analysis for the class of deep linear networks, demonstrating that as the horizon length approaches the total number of layers (or blocks), the gradient computed by the framework converges to that obtained using full BP. Additionally, numerical experiments validate the framework, offering both theoretical and practical insights into its performance across different models and tasks.

**Strengths:**

* The paper is overall well-written and clear.

* The approach of viewing deep neural network optimization through the lens of MPC is innovative and provides a fresh perspective.

* Additionally, the experiments and theoretical results are well-aligned and effectively complement each other, strengthening the overall argument of the paper.

**Weaknesses:**

* The paper overlooks a significant body of prior works that address the memory limitations of BP. Notable examples include techniques like checkpointing, forward-mode automatic differentiation, forward gradients [1]. I recommend that the authors include a comparison of their MPC framework with these memory-efficient techniques, specifically highlighting how their approach differs from or improves upon these existing methods in terms of memory savings and performance.

* Moreover, the time complexity of the proposed framework is not discussed. Based on my understanding, the time complexity would likely be $\mathcal{O}((T-h+1)h)$. For middle values of $h$, which the authors suggest might balance memory and performance, the time complexity actually increases by a factor of $\mathcal{O}(T)$. In this case, one could potentially use forward-accumulation gradients with the same time complexity and achieve better memory efficiency, while still producing gradients identical to BP (no performance loss). I suggest the authors provide a detailed analysis of the time complexity of their approach and clearly articulate the advantages of their framework compared to existing methods, particularly in terms of time and memory efficiency. This comparison would help clarify the specific benefits of the proposed approach over alternatives.

* A key experiment demonstrating the practical applicability of the framework is missing, particularly one that shows it can train a model from scratch with a small drop in performance while achieving significant memory savings. Without this, it is difficult to assess whether the proposed approach is useful in practice. I suggest the authors consider adding an experiment that compares training a model from scratch using their MPC framework (with various horizon lengths) against standard backpropagation, reporting both performance metrics and memory usage. This would provide concrete evidence of the framework's practical benefits and limitations.

[1] Baydin, Atılım Güneş, et al. "Gradients without backpropagation." arXiv preprint arXiv:2202.08587 (2022).

**Questions:**

* In Figure 3, what does "full tuning" refer to? Does this experiment involve training the models from scratch, or is it a fine-tuning process? I'm confused due to the use of "full tuning" in Figure 3 but "fine tuning" in Table D2.

* What is the significance of introducing the framework through MPC? Does it help with the analysis of the method? Given that intermediate terms cancel out in equation (6), the connection to MPC seems somewhat contrived and appears to introduce unnecessary complexity without providing clear benefits in understanding.

* Is the use of "max" in Objectives (1) and (2) a typo?

---

### Official Review · Reviewer_s82N · 2024-11-02

**Soundness:** 2
**Presentation:** 2
**Contribution:** 1
**Rating:** 3
**Confidence:** 4

**Summary:**

This paper tries to provide a unified training algorithm that connects BP and Forward-Forward algorithm based on the concepts or basic formulation in Model Predictive Control (MPC). The proposed training algorithm balances the accuracy and memory usage. The theoretical analysis is based on a deep linear model, followed by a horizon selection algorithm. Experiments are conducted by considering mang commonly used deep models.

**Strengths:**

The proposed training algorithm is technically sound, which interpolates between BP and FF.

The writing is clear and easy to follow.

**Weaknesses:**

1.Motivation. I understand it is a doable research paper to trade off memory usage and accuracy. However, I feel it may not be necessary to sacrifice accuracy to gain memory efficiency, given that nowadays, we have relatively sufficient computation power to train large deep models, e.g., foundation models. Thus, it may be less pressing to consider this trade-off.

2.Methodology. The proposed method simply borrows the concept of basic formulation of MPC without involving much technical content from MPC literature. Thus, I could not tell sufficient technical contribution in terms of methodology. Similarly, the title also seems misleading by emphasizing MPC too much.

3.Theory. One apparent limitation is the authors only derive results based on deep linear models, which could be fundamentally different from modern (non-linear) deep models, such as ResNet and Transformers. Although the heuristic extensions to modern deep models in experiments validate the theoretical results, this limitation is still non-neglectable.

4.Writing. The writing needs substantial improvement. Grammar errors include Line 35 (no subject) and Line 112 (not complete). The citation is also problematic, such as line 78.

5.The proposed method is motivated from a mere optimization perspective without considering the generalization or learning theory, which can be fundamentally limited. For example, it seems that Figure 4 only consider the training loss instead of looking into the test loss.

6.The choice of functions in Section 4 seems subjective, which is less convincing.

**Questions:**

Please see the weakness.

---

### Official Review · Reviewer_HB74 · 2024-11-04

**Soundness:** 2
**Presentation:** 2
**Contribution:** 2
**Rating:** 3
**Confidence:** 3

**Summary:**

Drawing inspiration from the model predictive control framework, this work proposes a framework for integrating back-propagation (BP) and the forward-forward (FF) algorithm (Hinton, 2022) for optimizing neural networks. In this framework, layer-wise local losses are back-propagated by $h$ steps, where the horizon $h$ is a user-provided algorithm parameter that controls a memory-performance trade-off. Here, $h=1$ corresponds to the FF algorithm, while $h=T$ for a $T$-layer neural net corresponds to backprop. A theoretical result is provided showing the convergence of the loss gradient to (a scaling of) the true gradient as $h \rightarrow T$. Assuming linear increase in memory consumption with $h$, the work also proposes a heuristic for selecting $h$ adaptively given a particular deep learning optimization problem, and hardware constraints or performance requirements. Empirical studies show the approach may be feasible for obtaining optimization algorithms that enable trading off performance for memory with more flexibility than FF, as well as another alternative (LoCo).

**Strengths:**

1. The paper offers a _fresh_ perspective, unifying BP and FF with inspiration from the MPC framework in search of a more flexible family of optimization algorithms.
2. The results obtained (both theoretical and empirical) show some promise in terms of controlling memory and performance trade-offs via the horizon parameter $h$.
3. Experimental setup is reasonably well-structured and conducive to conveying the main messages of the paper.

**Weaknesses:**

1. The connection to MPC seems like a bit of a stretch and makes the paper unnecessarily harder to digest in my opinion. That is, I found Sec. 3.2 to be needlessly long and dense; the same horizon idea could be described in simpler terms. The reason why I think the MPC connection is a bit of a stretch is that MPC applies _the optimal solution_ of the opt. problem to control the system given the trajectory cost, whereas the proposed approach takes a single gradient step.
2. In L211-214, the comments on memory usage read as though FF and/or the proposed framework has better _complexity_ than BP (re. usage of the word "growth"), when in fact the complexity is the same and gains are only in terms of constants. Indeed, Fig. 3 shows that FF ($h=1$) reduces memory by some factor of 3-4x in the best case and at a huge performance discount. Given modern hardware and distributed training capabilities, this brings to question whether interpolating FF and BP is worth the effort and complication to begin with (Occam's razor).
3. The theoretical result in Thm. 3.4 does not surprise me. Just looking at Fig. 1, one can already see that the gradients will be aligned exactly for roughly $h/T$ fraction of the parameters. Once again, I am not convinced the gravity of the result is worth the complication. Furthermore, the commentary in L270-271 seem to claim that alignment of the gradients necessarily translate to better performance, which I don't believe is true. Consider the Newton direction, which almost never aligns with the gradient, yet would likely yield much better performance than the gradient (steepest descent dir.) if it could be feasibly computed.
4. The horizon selection algorithm requires some runs with $h=T$. If this is possible on the available hardware, why bother reducing memory usage (except maybe some atypical use cases)?
5. Fig. 3 (right) is missing bars on memory usage, which seems awkward and raises suspicion for the reader. Note also that the linear memory demand assumption seems only to hold for eager execution (but not static execution) of the backprop framework. This information should be highlighted in the main text. Currently it's only mentioned in Appx. E.1.
6. The same goes for the range of values considered on the x-axis of Fig. 2. The scale for the rightmost 2 plots should also go down to $\approx 5 \times 10^{-3}$ like the leftmost plot.
7. Overreaching claims: e.g., L492 says "proposed horizon selection algorithm is _more efficient than_ BP". Careful wording is critical for maintaining scientific tone. Perhaps it's better to say something like "more memory-efficient than BP" or "better at optimizing our proposed objectives in (17-18)".

**Questions:**

1. It is not clear to me why alignment gets worse with more training (re. Fig. 2 and L400-401).
2. Suggestion: Fig. 5 would be easier to read if the caption included a note "lower is better".
3. Suggestion: Before introducing (19), referring the reader back to (17-18) might improve readability.
4. Suggestion: The use of the word "Objective" in the sense of (17-18) can be confusing for the reader, seeing as BP and FF also optimize losses (or, "objectives").

---

### Note · Authors · 2024-11-18

**Comment:**

We appreciate the reviewers’ time and constructive feedback. After further consideration, we have decided to withdraw the submission and will work on improving the paper based on the feedback provided

**Withdrawal Confirmation:**

I have read and agree with the venue's withdrawal policy on behalf of myself and my co-authors.